# Restricted Phase Space Thermodynamics of Einstein-Power-Yang–Mills AdS Black Hole

**DOI:** 10.3390/e25040687

**Published:** 2023-04-19

**Authors:** Yun-Zhi Du, Huai-Fan Li, Yang Zhang, Xiang-Nan Zhou, Jun-Xin Zhao

**Affiliations:** 1Institute of Theoretical Physics, Shanxi Datong University, Datong 037009, China; huaifan999@163.com (H.-F.L.); zhangyphysics@126.com (Y.Z.); 2College of Physics and Information Engineering, Shanxi Normal University, Taiyuan 030619, China; zhouxn10@let.edu.cn; 3China Citik Bank, Beijing 100000, China

**Keywords:** restricted phase space, phase transition, supercritical behaviour

## Abstract

We consider the thermodynamics of the Einstein-power-Yang–Mills AdS black holes in the context of the gauge-gravity duality. Under this framework, Newton’s gravitational constant and the cosmological constant are varied in the system. We rewrite the thermodynamic first law in a more extended form containing both the pressure and the central charge of the dual conformal field theory, i.e., the restricted phase transition formula. A novel phenomena arises: the dual quantity of pressure is the effective volume, not the geometric one. That leads to a new behavior of the Van de Waals-like phase transition for this system with the fixed central charge: the supercritical phase transition. From the Ehrenfest’s scheme perspective, we check out the second-order phase transition of the EPYM AdS black hole. Furthermore the effect of the non-linear Yang–Mills parameter on these thermodynamic properties is also investigated.

## 1. Introduction

Since black holes and their thermodynamics can provide clues about the nature of quantum gravity, they have been of crucial importance. Especially, the asymptotically anti-de Sitter (AdS) black holes with a finite temperature can provide a description of the dual conformal field theory (CFT) via the AdS/CFT correspondence [1]. Such black holes can be in thermal equilibrium with their radiation field and exhibit the Hawking–Page (HP) phase transition [2,3]. Subsequently, a very crucial point had been proposed: a negative cosmological constant can reduce a positive thermodynamic pressure, whose dual thermodynamic quantity is volume [4]. That makes AdS black holes identical to ordinary thermodynamic systems, and their thermodynamics become more complete. In this extended phase space, the mass parameter is interpreted as the entropy rather than the internal energy, and AdS black hole thermodynamics become richer and richer, such as the Van de Waals-like phase transition for the charged AdS black holes [5,6,7], the reentrant phase transitions for the rotating system [8,9], superfluid [10], the polymer-like phase transition [11], and the triple points [12,13], along with the novel dual relation of HP phase transition [14]. Meanwhile, the inclusion of the pressure–volume term in the thermodynamic first law makes other model parameters novel thermodynamic quantities [6] and makes it possible to regard AdS black holes as heat engines [15,16]. All of those developments are in the subdiscipline, black hole chemistry [17].

On this issue, people always attempt to give the concrete physical explanation of black holes chemistry in the extended phase space via AdS/CFT [18,19]. However, it is somewhat elusive from the viewpoint of the holographic [20,21]. For an AdS black hole system, the variations of the cosmological constant Λ correspond to both the changing of the central charge and the CFT volume, which indicates that the thermodynamic first law in the extended phase space cannot be straightforwardly related to the corresponding thermodynamics of the dual field theory [22,23]. Furthermore, the variation of pressure (or Λ) implies changing the gravity model. The corresponding ensemble does not describe the collection of black holes from the same gravity model where the macro states are the same, while it describes the collection of gravity models of the same or similar black hole solutions. Additionally, in the extended phase space, the absolute values of the coefficients appearing in the thermodynamic first law are not all one. These comments motivate the modification of the thermodynamic first law. Recently, the authors in Refs. [24,25] put forward the central charge and the chemical potential as a new pair of dual thermodynamic quantities that should be included in the thermodynamic first law. Here, Newton’s gravitational constant can change as well as Λ, which will induce profound consequences of the chemical potential and its holographic interpretation, and compared with ordinary thermodynamic systems, the introduction of these two thermodynamic quantities gives rise to a new thermodynamic phenomenon, the supercritical phase transition [26,27,28]. In this work, for the Einstein-power-Yang–Mills (EPYM) AdS black hole [29,30,31,32], we will exhibit the concrete process of establishing the more extended thermodynamic first law in which the pressure, volume, central charge, and chemical potential are included.

As we all know, at the linear level, the charged black holes in an AdS spacetime nearby the critical point is of the scaling symmetries, $S∼q^{2},\;P∼q^{-2},\;T∼q^{-1}$ [33,34], where *q* is the electric charge. Does the same scaling symmetry still hold for the non-linear charged AdS black holes? There are lots of generalizations of the linear-charged AdS black hole solution: Einstein–Maxwell–Yang–Mills AdS black hole [35], Einstein-power-Yang–Mills AdS black hole [32], Einstein–Maxwell-power-Yang–Mills AdS black hole [29], Einstein–Yang–Mills–Gauss–Bonnet black hole [36], Einstein-power-Maxwell-power-Yang–Mills dilaton [37], and so on. An interesting non-linear generalization of charged black holes involves a Yang–Mill field exponentially coupled to Einstein gravity (i.e., Einstein-power-Yang–Mills gravity theory) because it possesses the conformal invariance and is easy to construct the analogues of the four-dimensional Reissner–Nordström black hole solutions in higher dimensions. Additionally, several thermodynamic features of the EPYM AdS black hole in the extended phase space have been exhibited [29,38,39]. Do new thermodynamic phenomena appear for the EPYM AdS black hole when the central charge and the chemical potential are introduced? The answer will be given in this work.

This work is organized as follows. In Section 2, we briefly review the EPYM AdS black hole solution and its hawking temperature. In Section 3, we derive the more extended thermodynamic first law that includes the central charge and the chemical potential by considering the variation of Newton’s gravity constant and the cosmological constant. Then, critical thermodynamic quantities are exhibited, and the effect of the non-linear YM parameter on the critical point is also investigated in Section 4. In Section 5, we explore the first-order phase transition in this more extended phase space and compare the results with those in the extended phase space. Finally, we check out the Ehrenfest’s scheme of the EPYM AdS black hole in Section 6. A brief summary is given in Section 7.

## 2. EPYM AdS Black Hole and Hawking Temperature

The action for four-dimensional Einstein-power-Yang–Mills (EPYM) gravity with a cosmological constant Λ was given by [20,30,31,32]
(1)$$\begin{array}{c} I=\frac{1}{2}\int d^{4}x\sqrt{g}\left(R-2\Lambda -F^{\gamma }\right) \end{array}$$
with the Yang–Mills (YM) invariant F and the YM field $F_{\mu \nu }^{\left(a\right)}$
(2)$$\begin{array}{ccc} F & = & \mathrm{Tr}(F_{\mu \nu }^{\left(a\right)}F^{(a)\mu \nu }), \end{array}$$
(3)$$\begin{array}{ccc} F_{\mu \nu }^{\left(a\right)} & = & \partial_{\mu }A_{\nu }^{\left(a\right)}-\partial_{\nu }A_{\mu }^{\left(a\right)}+\frac{1}{2\xi }C_{\left(b)(c\right)}^{\left(a\right)}A_{\mu }^{\left(b\right)}A_{\nu }^{\left(c\right)}. \end{array}$$ Here, $\mathrm{Tr}\left(F_{\mu \nu }^{\left(a\right)}F^{(a)\mu \nu }\right)=\sum_{a=1}^{3}F_{\mu \nu }^{\left(a\right)}F^{(a)\mu \nu }$, *R* and γ are the scalar curvature and a positive real parameter, respectively; $C_{\left(b)(c\right)}^{\left(a\right)}$ represents the structure constants of three-parameter Lie group *G*; ξ is the coupling constant; and $A_{\mu }^{\left(a\right)}$ represents the SO(3) gauge group Yang–Mills (YM) potentials defined by the Wu–Yang (WY) ansatz [40]. Variation of the action with respect to the spacetime metric $g_{\mu \nu }$ yields the field equations
(4)$$\begin{array}{ccc} & & G^{\mu }{}_{\nu }+\Lambda \delta^{\mu }{}_{\nu }=T^{\mu }{}_{\nu }, \end{array}$$
(5)$$\begin{array}{ccc} & & T^{\mu }{}_{\nu }=-\frac{1}{2}\left(\delta^{\mu }{}_{\nu }F^{\gamma }-4\gamma \mathrm{Tr}\left(F_{\nu \lambda }^{\left(a\right)}F^{(a)\mu \lambda }\right)F^{\gamma -1}\right). \end{array}$$ Variation with respect to the 1-form YM gauge potentials $A_{\mu }^{\left(a\right)}$ and implementing the traceless yields the 2-forms YM equations
(6)$$d\left({}^{★}F^{\left(a\right)}F^{\gamma -1}\right)+\frac{1}{\xi }C_{\left(b)(c\right)}^{\left(a\right)}F^{\gamma -1}A^{\left(b\right)}\wedge^{★}F^{\left(c\right)}=0,$$
where $F^{\left(a\right)}=\frac{1}{2}F_{\mu \nu }^{\left(a\right)}dx^{\mu }\wedge dx^{\nu },\;A^{\left(b\right)}=A_{\mu }^{\left(b\right)}\wedge dx^{\mu }$, and ${}^{★}$ stands for the duality. It is obvious that for the case of γ=1 the EPYM theory reduces to the standard Einstein–Yang–Mills (EYM) theory [36]. In this work, our issue is paid on the role of the non-linear YM charge parameter γ.

Here, we should point out that the non-Abelian property of the YM gauge field is expressed with its YM potentials
(7)$$\begin{array}{c} A^{\left(b\right)}=\frac{q}{r^{2}}C_{\left(i)(j\right)}^{\left(a\right)}x^{i}dx^{j},\;r^{2}=\sum_{j=1}^{3}x_{j}^{2}, \end{array}$$
and *q* is the YM charge, and the indices (a,i,j) run the following ranges: 1≤a,i,j≤3. The coordinates $x_{i}$ take the following forms: $x_{1}=r\cos\phi \sin\theta ,\;x_{2}=r\sin\phi \sin\theta ,\;x_{3}=r\cos\theta .$ Since we have utilized the WY ansatz for the YM field, the invariant for this field takes the form [41,42]
(8)$$\begin{array}{c} \mathrm{Tr}\left(F_{\mu \nu }^{\left(a\right)}F^{(a)\mu \nu }\right)=\frac{q^{2}}{r^{4}}. \end{array}$$ This form leads to the disappearance of the structure constants which can be described by the non-Abelian property of the YM gauge field. Therefore, under the condition of the WY ansatz, we may focus on the role of the non-linear YM charge parameter instead of the non-Abelian character parameter.

The metric for the four-dimensional EPYM AdS black hole is given as follows [43],
(9)$$\begin{array}{c} ds^{2}=-f\left(r\right)dt^{2}+f^{-1}dr^{2}+r^{2}d\Omega_{2}^{2}, \end{array}$$
where
(10)$$\begin{array}{c} f\left(r\right)=1-\frac{2GM}{r}+\frac{r^{2}}{l^{2}}+\frac{G{\left(2q^{2}\right)}^{\gamma }}{2\left(4\gamma -3\right)r^{4\gamma -2}}. \end{array}$$ Here, $d\Omega_{2}^{2}$ is the metric on the two-sphere unit with volume fourπ, and *q* is the YM charge, *l* is related to the cosmological constant: $l^{2}=-\frac{3}{\Lambda }$, and γ is the non-linear YM charge parameter and satisfies γ>0 [30]. The event horizon of the black hole is obtained from the relation $f(r_{+})=0$. There exist two roots of the relation f(r)=0. One is the inner horizon $r_{-}$. Another is the outer horizon $r_{+}$. Generally, the AdS black hole event horizon means the outer horizon. The mass parameter of the black hole can be expressed in terms of the horizon radius as
(11)$$\begin{array}{c} M=\frac{r_{+}}{2G}\left(1+\frac{r_{+}^{2}}{l^{2}}+\frac{2^{\gamma -1}Gq^{2\gamma }}{\left(4\gamma -3\right)r_{+}^{4\gamma -2}}\right). \end{array}$$ We can also obtain the Hawking temperature of the black hole from Equation (Equation 10) as follows
(12)$$\begin{array}{c} T=\frac{1}{4\pi r_{+}}\left(1+8\pi GPr_{+}^{2}-\frac{G{\left(2q^{2}\right)}^{\gamma }}{2r_{+}^{\left(4\gamma -2\right)}}\right). \end{array}$$ From Equations (Equation 11) and (Equation 12), we will calculate the critical value of thermodynamic quantities which are presented in Section 6. Next, we will give the modified first law of the four-dimensional EPYM AdS black hole thermodynamics in natural units (ℏ=c=1), i.e., the restricted phase space formulism.

## 3. Restricted Phase Space Formulism of EPYM AdS Black Hole

Recently, people in Refs. [4,5] proposed that the negative cosmological constant could induce a positive thermodynamic pressure, which is in terms of the cosmological constant and Newton’s gravitational constant as
(13)$$\begin{array}{c} P=-\frac{\Lambda }{8\pi G}\;\;\;\;\mathrm{or}\;\;\;\;P=\frac{3}{8\pi Gl^{2}}. \end{array}$$ In the above equation, the pressure will change with the variation of the cosmological constant and Newton’s gravitational constant. In natural units, the Bekenstein–Hawking entropy reads
(14)$$\begin{array}{c} S=\frac{A}{4G}=\frac{\pi r_{+}^{2}}{G}, \end{array}$$
where *A* is the area of the black hole. The Hawking temperature can be expressed in terms of the surface gravity κ as
(15)$$\begin{array}{c} T=\frac{\kappa }{2\pi }. \end{array}$$ In the extended phase space, Newton’s gravitational constant is fixed, and the mass of the black hole is interpreted as the enthalpy instead of the internal energy. Thus, the general form of the thermodynamic first law for the EPYM AdS black hole of the surface gravity, charge, the cosmological constant, and the area are
(16)$$\begin{array}{ccc} \delta M & = & T\delta S+V\delta P+\Psi \delta q^{2\gamma }\\ & = & \frac{\kappa }{8\pi G}\delta A-\frac{V}{8\pi G}\delta \Lambda +\Psi \delta q^{2\gamma }, \end{array}$$
where the volume and potential are
(17)$$\begin{array}{c} V=\frac{4\pi r_{+}^{3}}{3},\;\;\;\;\Psi =\frac{2^{\gamma -2}}{\left(4\gamma -3\right)r_{+}^{4\gamma -3}}. \end{array}$$ We can check out the final expression in Equation (Equation 16) by using Equations (Equation 13)–(Equation 15). In the expanded phase space, the thermodynamic phase transition properties of the EPYM AdS black hole were exhibited in Ref. [38] and the corresponding optical properties including the photon sphere and shadow were also presented in Refs. [44,45]. Next, we will exhibit the concrete details of the restricted phase space formulism for the EPYM AdS black hole in the case of γ≠3/4.

It has been shown that the holographic interpretation of the above thermodynamic first law in Equation (Equation 16) could cause some issues [15,46,47]. The VδP (the variation of the cosmological constant) in the thermodynamic first law of the bulk is shown to have two terms in the thermodynamic first law at the boundary conformal field theory (CFT): one is the central charge of the boundary CFT and the thermodynamic pressure of the boundary CFT which is caused by the change of the AdS radius. The way of addressing this problem is to invoke the form of the central charge from the AdS/CFT dictionary, which is related to the AdS radius *l* as in Ref. [22]
(18)$$\begin{array}{c} C=\frac{kl^{2}}{16\pi G}. \end{array}$$ Here, the parameter *k* is determined on the details of the system on the boundary. Note that the first law of thermodynamics in the extended phase space (see Equation (Equation 16)) cannot be straightforwardly related to the corresponding thermodynamics of the holographic dual field theory because variations of the bulk cosmological constant correspond to changing both the central charge and the CFT volume. Indeed, it also corresponds to changing the notion of electric charge and the corresponding potential that both rescale with the AdS radius. It is possible to hold the central charge fixed so that the field theory remains the same by simultaneously varying Newton’s constant. Here, we demonstrate that the variation of Newton’s constant has profound consequences for black hole chemistry and its holographic interpretation. We build on the previous holographic generalizations of the first law, Equation (Equation 16), which include variations of Newton’s constant by rewriting it in a new mixed form, in terms of variations of the cosmological constant and the central charge. The explicit appearance of δC allows for a study of bulk thermodynamics in the same CFT theory on the AdS boundary and yields a new definition for the thermodynamic black hole volume. When considering the mass parameter *M* to be a function of the area *A*, the cosmological constant Λ, the charge $q^{2\gamma }$, and Newton’s gravitational constant *G*, i.e., $M\equiv M(A,\;\Lambda ,\;q^{2\gamma },\;G)$, the variation of *M* can be rewritten as
(19)$$\begin{array}{c} \delta M=\frac{\partial M}{\partial A}\delta A+\frac{\partial M}{\partial \Lambda }\delta \Lambda +\frac{\partial M}{\partial q^{2\gamma }}\delta q^{2\gamma }+\frac{\partial M}{\partial G}\delta G. \end{array}$$ Compared with Equation (Equation 16), we can see that the conjugate variables of *A*, Λ, and $q^{2\gamma }$ are $\frac{\kappa }{8\pi G},\;-\frac{V}{8\pi G},$ and Ψ. With the definition $G\frac{\partial M}{\partial G}=-\xi$, we can recast the above equation as
(20)$$\begin{array}{c} \delta M=\frac{\kappa }{8\pi G}\delta A-\frac{V}{8\pi G}\delta \Lambda +\Psi \delta q^{2\gamma }-\xi \frac{\delta G}{G}. \end{array}$$ In the following, we try to give the coefficient ξ in the above equation. For that, we make use of a modified mass term as suggested in [24]
(21)$$\begin{array}{c} GM=M(A,\Lambda ,Gq^{2\gamma }). \end{array}$$ Performing the differential of the above equation and combining Equation (Equation 19), we have
(22)$$\begin{array}{ccc} G\delta M & = & \frac{\partial M}{\partial A}\delta A+\frac{\partial M}{\partial \Lambda }\delta \Lambda +\frac{G\partial M}{\partial (Gq^{2\gamma })}\delta q^{2\gamma }+\left(\frac{q^{2\gamma }\partial M}{\partial (Gq^{2\gamma })}-M\right)\delta G, \end{array}$$
(23)$$\begin{array}{ccc} \Rightarrow \;\;\;\;\delta M & = & \frac{\partial M}{G\partial A}\delta A+\frac{\partial M}{G\partial \Lambda }\delta \Lambda +\frac{\partial M}{\partial (Gq^{2\gamma })}\delta q^{2\gamma }+\left(\frac{q^{2\gamma }\partial M}{\partial (Gq^{2\gamma })}-M\right)\frac{\delta G}{G}. \end{array}$$ Comparing the above equation with Equation (Equation 20), we can obtain the following expressions
(24)$$\begin{array}{c} \frac{\partial M}{\partial A}=\frac{\kappa }{8\pi },\;\;\;\;\frac{\partial M}{\partial \Lambda }=-\frac{V}{8\pi },\;\;\;\;\frac{\partial M}{\partial (Gq^{2\gamma })}=\Psi ,\;\;\;\;\frac{q^{2\gamma }\partial M}{\partial (Gq^{2\gamma })}-M=-\xi . \end{array}$$ Therefore, the coefficient ξ has the form as
(25)$$\begin{array}{c} \xi =M-q^{2\gamma }\Psi . \end{array}$$ Performing the differential of the cosmological constant, the pressure in Equation (Equation 13), the area in Equation (Equation 14), and the central charge in Equation (Equation 18), we have
(26)$$\begin{array}{c} \frac{\delta \Lambda }{\Lambda }=-\frac{\delta C}{2C}+\frac{\delta P}{2P},\;\;\frac{\delta G}{G}=-\left(\frac{\delta C}{2C}+\frac{\delta P}{2P}\right),\;\;\frac{\delta A}{A}=\frac{\delta S}{S}-\frac{\delta P}{2P}-\frac{\delta C}{2C}, \end{array}$$ Combining these results, the differential of the black hole mass in Equation (Equation 23) can be rewritten as a function of the thermodynamic quantities ($S,\;P,\;q^{2\gamma },\;C$)
(27)$$\begin{array}{c} \delta M=T\delta S+V_{eff}\delta P+\Psi \delta q^{2\gamma }+\mu \delta C, \end{array}$$
where $V_{eff}$ and μ are the effective thermodynamic volume and the chemical potential, and they have the following forms
(28)$$\begin{array}{ccc} V_{eff} & = & \frac{1}{2P}\left(M-TS+PV-q^{2\gamma }\Psi \right), \end{array}$$
(29)$$\begin{array}{ccc} \mu & = & \frac{1}{2C}\left(M-TS-PV-q^{2\gamma }\Psi \right). \end{array}$$ That is the restricted phase space formulism of the four-dimensional EPYM AdS black hole in the case of γ≠3/4. For convenience, we introduce the induced thermodynamic quantities as
(30)$$\begin{array}{c} \bar{M}=GM,\;\;\bar{S}=GS,\;\;\bar{P}=GP,\;\;\bar{q}=Gq^{2\gamma },\;\;\bar{C}=GC,\;\; \end{array}$$
the thermodynamic first law in the restricted phase space becomes
(31)$$\begin{array}{c} \delta \bar{M}=T\delta \bar{S}+V_{eff}\delta \bar{P}+\Psi \delta \bar{q}+\mu \delta \bar{C}, \end{array}$$
and the effective volume and the chemical potential are
(32)$$\begin{array}{ccc} V_{eff} & = & \frac{1}{2\bar{P}}\left(\bar{M}-T\bar{S}+\bar{P}V-\frac{\bar{q}\Psi }{2}\right), \end{array}$$
(33)$$\begin{array}{ccc} \mu & = & \frac{1}{2\bar{C}}\left(\bar{M}-T\bar{S}-\bar{P}V-\frac{\bar{q}\Psi }{2}\right). \end{array}$$ From the above equations and Equation (Equation 17), we can see that the Euler relation of EPYM AdS black holes in the restricted phase space is indeed restored as in an ordinary thermodynamic system, which reads
(34)$$\begin{array}{c} \bar{M}=T\bar{S}+\bar{P}V_{eff}+\bar{q}\Psi +\mu \bar{C}. \end{array}$$ In the following, we will use these induced quantities ($\bar{S},\;\bar{P},\;\bar{q},\;\bar{C}$) and $T,\;V_{eff},\;\Psi ,\;\mu$ to investigate the thermodynamic properties of this system.

## 4. Critical Curves of EPYM AdS Black Hole

Based on the classification of phase transition for a thermodynamic system by Ehrenfest, the critical point can be obtained by the following equations
(35)$$\begin{array}{c} \frac{\partial T}{\partial \bar{S}}=\frac{\partial^{2}T}{\partial {\bar{S}}^{2}}=0. \end{array}$$ With Equations (Equation 12) and (Equation 14) and the above equations, we have
(36)$$\begin{array}{c} r_{c}^{4\gamma -2}=\gamma \left(4\gamma -1\right)2^{\gamma }\bar{q},\;\;\;\;l_{c}^{2}=\frac{6\gamma }{2\gamma -1}r_{c}^{2}. \end{array}$$ The other critical quantities in the restricted phase space are
(37)$$\begin{array}{ccc} {\bar{P}}_{c} & = & \frac{2\gamma -1}{16\pi \gamma r_{c}^{2}},\;\;V_{eff}^{c}=\frac{1}{3}\pi r_{c}^{3}\left(\frac{6\gamma }{2\gamma -1}+\frac{3\gamma 2^{\gamma +1}\bar{q}r_{c}^{2-4\gamma }}{4\gamma -3}+1\right), \end{array}$$
(38)$$\begin{array}{ccc} T_{c} & = & \frac{2\gamma -1}{\left(4\gamma -1\right)\pi r_{c}},\;\;{\bar{S}}_{c}=\pi r_{c}^{2}, \end{array}$$
(39)$$\begin{array}{ccc} {\bar{C}}_{c} & = & \frac{3\gamma r_{c}^{2}}{8\pi (2\gamma -1)},\;\;\mu_{c}=\frac{\pi (2\gamma -1)}{6\gamma^{2}r_{c}}\left(\frac{2^{\gamma +1}\gamma \left(2\gamma -1\right)\bar{q}r_{c}^{2-4\gamma }}{4\gamma -3}+1\right) \end{array}$$ The results indicate that the critical point is determined by the non-linear YM parameter γ and the YM charge $\bar{q}$. Since the effect of $\bar{q}$ on the phase transition has been investigated in previous works [38,44], here we only exhibit the effects of γ on the critical temperature, the critical pressure, and the critical central charge in Figure 1. In the range 0.5<γ≤0.5982, the critical temperature and pressure are both decreasing with the increase of γ, while the critical central charge is increasing. When 0.6456≤γ, the critical temperature and pressure are the monotonically increasing functions with γ, while the critical central charge is not. In the middle range 0.5982≤γ≤0.6456, the critical central charge and pressure is increasing, while the critical temperature is decreasing.

## 5. First-Order Phase Transition in Restricted Phase Space

In previous works [38,44], the phase transition condition of the EPYM AdS black hole in the extended phase space was proposed. Furthermore, the first-order phase diagrams of T−S, P−V, and $q^{2\gamma }-\Psi$ in the extended phase space were also exhibited. In this manuscript, we introduce the central charge and the chemical potential to present the phase structure of the EPYM AdS black hole in the restricted phase space, where the volume is modified. Thus, we mainly focus on the phase diagrams in the $T-\bar{S}$, $\bar{P}-V_{eff}$, and $\bar{C}-\mu$ planes and exhibit the corresponding properties of phase transition. First, we will review the phase transition condition from the viewpoint of the independent dual thermodynamic quantities $T-\bar{S}$.

For the EPYM black hole to the given YM charge $\bar{q}$ and pressure ${\bar{P}}_{0}<{\bar{P}}_{c}$ in the phase diagram of $T-\bar{S}$, the entropies at the boundary of the two-phase coexistence area are marked by ${\bar{S}}_{1}$ and ${\bar{S}}_{2}$, respectively. The corresponding phase transition temperature is $T_{0}$, which is related with the horizon radius $r_{+}$. Therefore, from the Maxwell’s equal-area law $T_{0}\left({\bar{S}}_{2}-{\bar{S}}_{1}\right)=\int_{{\bar{S}}_{1}}^{{\bar{S}}_{2}}Td\bar{S}$ and Equation (Equation 12), we have
(40)$$\begin{array}{c} 2\pi T_{0}=\frac{1}{r_{2}\left(1+x\right)}+\frac{8\pi {\bar{P}}_{0}r_{2}}{3(1+x)}\left(1+x+x^{2}\right)-\frac{2^{\gamma }\bar{q}r_{2}^{1-4\gamma }}{2(3-4\gamma )}\frac{\left(1-x^{3-4\gamma }\right)}{\left(1-x^{2}\right)} \end{array}$$
with $x=\frac{r_{1}}{r_{2}}$. In addition, from the state equation we have
(41)$$\begin{array}{c} T_{0}=\frac{1}{4\pi r_{1,2}}\left(1+8\pi {\bar{P}}_{0}r_{1,2}^{2}-\frac{2^{\gamma }\bar{q}}{2r_{1,2}^{\left(4\gamma -2\right)}}\right),\;\;\;\; \end{array}$$ From the two above equations, we have
(42)$$\begin{array}{ccc} 0 & = & -\frac{1-x}{r_{2}x}+8\pi {\bar{P}}_{0}r_{2}\left(1-x\right)+\frac{2^{\gamma }\bar{q}}{2r_{2}^{4\gamma -1}x^{4\gamma -1}}\left(1-x^{4\gamma -1}\right),\;\;\;\; \end{array}$$
(43)$$\begin{array}{ccc} 8\pi T_{0} & = & \frac{1+x}{r_{2}x}+8\pi {\bar{P}}_{0}r_{2}\left(1+x\right)-\frac{2^{\gamma }\bar{q}}{2r_{2}^{4\gamma -1}x^{4\gamma -1}}\left(1+x^{4\gamma -1}\right). \end{array}$$ Considering Equations (Equation 40), (Equation 42), and (Equation 43), the horizon $r_{2}$ has the following form
(44)$$\begin{array}{c} r_{2}^{4\gamma -2}=\frac{2^{\gamma }\bar{q}\left[\left(3-4\gamma \right)\left(1+x\right)\left(1-x^{4\gamma }\right)+8\gamma x^{2}\left(1-x^{4\gamma -3}\right)\right]}{2x^{4\gamma -2}\left(3-4\gamma \right){\left(1-x\right)}^{3}}=2^{\gamma }\bar{q}f\left(x,\gamma \right). \end{array}$$

Since the horizon radius must be positive, the non-linear YM charge parameter satisfies the condition $\frac{1}{2}<\gamma$ and $\gamma \neq \frac{3}{4}$. In addition, from the state Equation (Equation 41), the temperature $T_{0}$ can be written as a function of $r_{+}$ and *x*
(45)$$\begin{array}{c} T_{0}=\frac{1}{4\pi x\left(1-x\right)r_{2}}\left(1-x^{2}-\frac{2^{\gamma -1}\bar{q}\left(1-x^{4\gamma }\right)}{{\left(xr_{2}\right)}^{4\gamma -2}}\right). \end{array}$$ Considering Equations (Equation 44) and (Equation 45), we find that for the given values of γ, $\bar{q}$, and temperature $T_{0}$, we can calculate the value of *x*. Then, substituting the result of *x* into Equation (Equation 44), the large horizon radius will be obtained. Thus for the given temperature $T_{0}$ ($T_{0}<T_{c}$), the first-order phase transition condition reads
(46)$$\begin{array}{c} \frac{{\left(2q^{2}\right)}^{\gamma }}{r_{2}^{4\gamma -2}}=\frac{1}{f(x,\gamma )}. \end{array}$$ In other words, the phase transition of this system is determined by the ratio between the YM charge $\bar{q}$ and $r_{2}^{4\gamma -2}$, not just the black hole horizon. Note that we call this ratio the YM potential at the horizon surface $r_{2}$. Thus, we call this phase transition the high/low-potential black hole (HPBH/LPBL) one.

The phase diagrams of $\bar{P}-V_{eff}$, $\bar{C}-\mu$, and $T-\bar{S}$ are shown in Figure 2 and Figure 3a. It is very interesting that for the system with the higher central charge ($\bar{C}>{\bar{C}}_{c}$), the first-order phase transition appears, while it will vanish as $\bar{C}<{\bar{C}}_{c}$. This phenomena is consistent with that in Ref. [24] which is called the supercritical phase transition and governed by the freedom degree in conformal field theory. However it is completely different from the system undergoing the isobaric processes in the extended phase space [38] as well as another kind of supercritical phase transition [26] where the central charge and chemistry potential exist but not the pressure and volume. Therefore, it can be guessed that the effects of pressure and central charge on the first-order phase transition are completely opposite. In addition, although the volume in the restricted phase space is modified compared with that in the expanded phase space, the phase transition point is still the same in the volume–pressure plane. For the lower temperature (less than the critical one), there exists the first-order phase transition both in the $\bar{P}-V_{eff}$ and $\bar{C}-\mu$ planes. For the higher temperature, the first-order phase transition of the system disappears. However, there exists a very interesting phenomena: when the central charge of this system is lower than the critical one, the phase transition does not exist, while for the higher central charge it appears. That is completely different from other thermodynamic quantities. In addition, the effect of the non-linear YM parameter on the phase diagram of $\bar{P}-V_{eff}$ is shown in Figure 3b.

## 6. Phase Transition from Ehrenfest’s Equations

We now exploit Ehrenfest’s scheme in order to understand the nature of the phase transition. Ehrenfest’s scheme basically consists of a pair of equations known as Ehrenfest’s equations of the first and second kind. For a standard thermodynamic system, these equations may be written as [48,49,50]
(47)$$\begin{array}{ccc} {\left(\frac{\partial \bar{P}}{\partial T}\right)}_{\bar{S}} & = & \frac{C_{{\bar{P}}_{2}}-C_{{\bar{P}}_{1}}}{TV_{eff}\left(\beta_{2}-\beta_{1}\right)}=\frac{\Delta C_{\bar{P}}}{TV_{eff}\Delta \beta }, \end{array}$$
(48)$$\begin{array}{ccc} {\left(\frac{\partial \bar{P}}{\partial T}\right)}_{V_{eff}} & = & \frac{\beta_{2}-\beta_{1}}{\kappa_{T_{2}}-\kappa_{T_{1}}}=\frac{\Delta \beta }{\Delta \kappa_{T}}. \end{array}$$
where $\beta =\frac{1}{V_{eff}}{\left(\frac{\partial V_{eff}}{\partial T}\right)}_{\bar{P}}$ is the volume expansion coefficient, and $\kappa_{T}=-\frac{1}{V_{eff}}{\left(\frac{\partial V_{eff}}{\partial \bar{P}}\right)}_{T}$ is the isothermal compressibility coefficient. For a genuine second order phase transition, both of these equations have to be satisfied simultaneously. From Equation (Equation 31), we can obtain the following relation
(49)$${\left(\frac{\partial \bar{P}}{\partial T}\right)}_{\bar{S}}={\left(\frac{\partial \bar{S}}{\partial V_{eff}}\right)}_{\bar{P}},\;{\left(\frac{\partial \bar{P}}{\partial T}\right)}_{V_{eff}}={\left(\frac{\partial \bar{S}}{\partial V_{eff}}\right)}_{T}.$$ With the above equations, the Prigogine–Defay (PD) ratio Π becomes
(50)$$\Pi ={\left(\frac{\partial \bar{P}}{\partial T}\right)}_{\bar{S}}/{\left(\frac{\partial \bar{P}}{\partial T}\right)}_{V_{eff}}={\left(\frac{\partial \bar{S}}{\partial V_{eff}}\right)}_{\bar{P}}/{\left(\frac{\partial \bar{S}}{\partial V_{eff}}\right)}_{T}.$$ The definition of the PD ratio was presented by Prigogine and Defay [51] and reviewed in Ref. [52]. At the critical point $\left(T_{c},{\bar{P}}_{c},V_{eff}^{c}\right)$, we have
(51)$${\left(\frac{\partial \bar{P}}{\partial V_{eff}}\right)}_{T}={\left(\frac{\partial^{2}\bar{P}}{\partial V_{eff}^{2}}\right)}_{T}=0.$$ Substituting Equation (Equation 49) into Equations (Equation 47) and (Equation 48), at the critical point we can obtain
(52)$$\frac{\Delta C_{\bar{P}}}{T_{c}V_{eff}^{c}\Delta \beta }={\left[{\left(\frac{\partial \bar{S}}{\partial V_{eff}}\right)}_{\bar{P}}\right]}^{c},\;\frac{\Delta \beta }{\Delta \kappa_{T}}={\left[{\left(\frac{\partial \bar{S}}{\partial V_{eff}}\right)}_{T}\right]}^{c}.$$ On the other hand, since $\bar{S}=\bar{S}\left(\bar{P},\;V_{eff}\right)$, therefore,
(53)$$\begin{array}{c} {\left(\frac{\partial \bar{S}}{\partial V_{eff}}\right)}_{T}={\left(\frac{\partial \bar{S}}{\partial \bar{P}}\right)}_{V_{eff}}{\left(\frac{\partial \bar{P}}{\partial V_{eff}}\right)}_{T}+{\left(\frac{\partial \bar{S}}{\partial V_{eff}}\right)}_{\bar{P}}. \end{array}$$ From Equation (Equation 51), we can find that, ${\left(\frac{\partial \bar{P}}{\partial V_{eff}}\right)}_{T}=0$ and ${\left(\frac{\partial \bar{S}}{\partial \bar{P}}\right)}_{V_{eff}}$ have a finite value at the critical point. Therefore, the first term of the right side for the above equation vanishes. That is a very special thermodynamic feature of AdS black holes, which may not still hold for other systems. Thus, we have
(54)$$\begin{array}{c} {\left[{\left(\frac{\partial \bar{S}}{\partial V_{eff}}\right)}_{T}\right]}^{c}={\left[{\left(\frac{\partial \bar{S}}{\partial V_{eff}}\right)}_{\bar{P}}\right]}^{c}, \end{array}$$ Then, substituting Equation (Equation 54) into Equation (Equation 50), the universal PD ratio (∏) at the critical point becomes
(55)∏=1. Hence, from the PD ratio perspective, the phase transition occurring at $T=T_{c}$ is a second-order equilibrium transition as well as other AdS black holes [48,49,50]. This is also consistent with the results in the last section. In other words, the phase transition of AdS black holes is independent of the phase spaces, such as the extended phase space and the restricted phase space.

## 7. Discussions and Conclusions

In this manuscript, we studied the thermodynamics of the EPYM AdS black hole in the restricted phase space, which revealed several remarkable characteristics that are the same as the RN-AdS black hole, and compared them with those in the expanded phase space. The results are summarized in the following

The first law of thermodynamics for the EPYM AdS black hole in the restricted phase space conforms to the standard description of ordinary thermodynamic systems: the mass parameter is to be understood as the internal energy, and the Euler relation of this system in the restricted phase space is restored as in an ordinary thermodynamic system.In these two different phase spaces, the property of phase transition including the first-order and second-order phase transitions for the EPYM AdS black hole does not change. That means that the thermodynamic property of AdS black holes is independent of the adoption of corresponding phase spaces.From the PD ratio perspective, this charged non-linear black hole is indeed in an equilibrium state at $T=T_{c}$ as well as ordinary thermodynamic systems. This also indicates that black holes can be indeed regarded as thermodynamic systems.

## Figures and Tables

**Figure 1 entropy-25-00687-f001:**
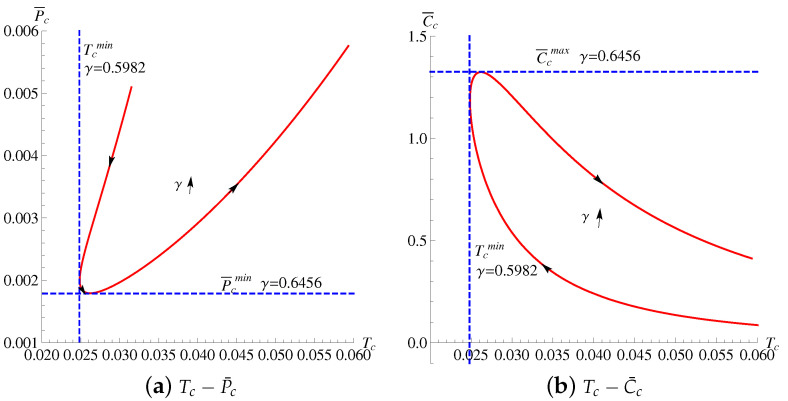
The critical curves in $T_{c}-{\bar{P}}_{c}$ and $T_{c}-{\bar{C}}_{c}$ diagrams with the non-linear charge parameter γ. The YM charge is set to $\bar{q}=1$.

**Figure 2 entropy-25-00687-f002:**
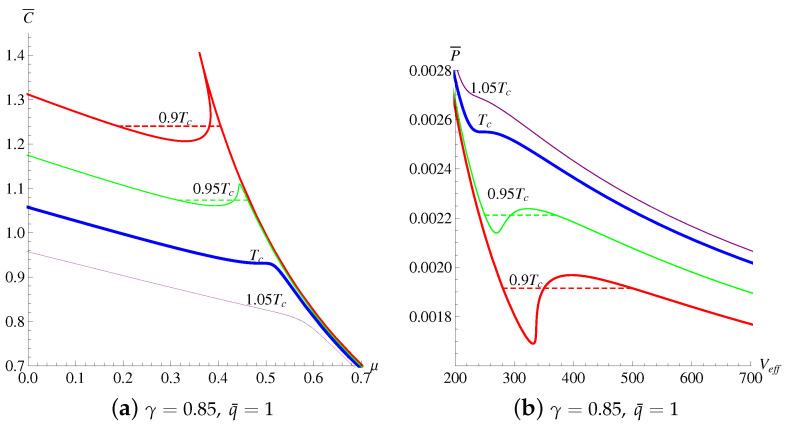
The phase diagrams of $V_{eff}-\bar{P}$ and $\mu -\bar{C}$ with different values of temperature.

**Figure 3 entropy-25-00687-f003:**
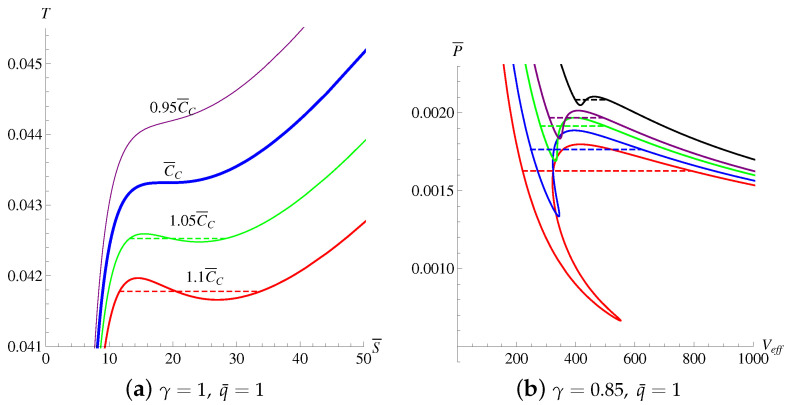
The first-order phase diagrams of $V_{eff}-\bar{P}$ with different values of temperature. The parameters are set to $\bar{q}=1,\;T=0.033<T_{c}$, and the non-linear YM charge parameter varies from 0.8 to 1 from the black line to the red one.

## Data Availability

There was no data, since this paper is a theoretical paper.

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
