# Peer review of "Restricted Phase Space Thermodynamics of Einstein-Power-Yang–Mills AdS Black Hole"

_entropy, 2023, doi:10.3390/e25040687_

Round 1
Reviewer 1 Report
This manuscript investigated the thermodynamics of the Einstein-Power-Yang-Mills AdS black holes as well as the Van de Waals-like phase transition. To understand the thermodynamics in the context of gauge/gravity duality, one expects that the Newton's constant and the cosmological constant are treated as variables in the system,then the dual quantity of pressure is restricted to the effective volume rather than the ordinary geometric one. A kind of supercritical phase transition is observed following the Ehrenfest's scheme. Meanwhile, I think the manuscript must be improved dramatically and the following points should be clarified before being considered for publication:
1. The main purpose of extending the first law of thermodynamics in Eq.(16) is based on the holographic consideration, as the authors claimed. To keep the central charge to be constant, one forces the Newton's ``constant" to vary with the AdS radius l, which sounds very peculiar. In particular, it requires that G is exactly proportional to the square root of the radius l, from Eq.(18). As a matter of fact, one could keep the central charge to be constant simply by fixing both the Newton's constant as well as the AdS radius l, as usually done in any holographic model. One should clarify this confusing point in the revised version. In addition, if the central charge is fixed to be constant, how could one understand its variation as appearing in the last term of (27)?
2.For the metric with f(r) in Eq.(10), it seems there exist more than one root for f(r)=0. Some details on the horizon $r_+$ should be presented.
3.Some typos should be corrected in the manuscript, such as ``a equilibrium state" in the last paragraph etc.
Author Response
First of all, we would like to express our sincere appreciation to the referee for his/her valuable time and efforts in evaluating our manuscript. Based on the comments and requests of the referee, we have made extensive modifications on the original manuscript. A detailed, point-to-point response is presented below.
Answer1: So sorry for the unclear exclaim on the central charge. Here we should point out that the first law of the thermodynamics in the extended phase space cannot be straightforwardly related to the corresponding thermodynamics of the holographic dual field theory because variations of the bulk cosmological constant correspond to changing both the central charge and the CFT volume. Indeed, it also corresponds to changing the notion of electric charge and the corresponding potential that both rescale with the AdS radius. It is possible to hold the central charge fixed so that the field theory remains the same by simultaneously varying Newton's constant. Here, we demonstrate that the variation of the Newton's constant has profound consequences for black hole chemistry and its holographic interpretation. We build on the previous holographic generalizations of the first law, eq. (16), which include variations of the Newton's constant by rewriting it in a new mixed form, in terms of variations of the cosmological constant and the central charge. The explicit appearance of δC allows for a study of bulk thermodynamics in the same CFT theory on the AdS boundary and yields a new definition for the thermodynamic black hole volume. The corresponding comments of the variation of the central charge in the restricted phase space have been added in the modified manuscript.
Answer2: At this point, we give the following comments. There exist two roots of the relation f(r)=0, one is the inner horizon r-, another is the outer horizon r+. Generally, the AdS black hole event horizon means the outer horizon. The corresponding comments have been added in the modified manuscript.
Answer3: Thank you for your suggestion. "a equilibrium state" had been corrected as "an equilibrium state".

Reviewer 2 Report
Entropy-2318728
In this manuscript the authors describe the thermodynamics of the EPYM AdS black hole in the restricted phase space. The authors compare the thermodynamical properties in both the restricted and expanded phase spaces and found that several characteristics are equivalent. In particular, the authors found that some thermodynamical properties of AdS black holes are independent on the phase space.
The authors also explore Ehrenfest's scheme in order to understand the nature of the phase transition.
The paper is well written and the procedure is well developed. I thus consider the manuscript is suitable for publication in Entropy its present form.
Author Response
Thank you very much for your kindness review and affirmation of our work. Some mistakes of English language have been modified in our manuscript.
Round 2
Reviewer 1 Report
The authors have partially answered my questions and improved the manuscript. I recommend it for publication.